# A Phase I/II randomized trial of H56:IC31 vaccination and adjunctive cyclooxygenase-2-inhibitor treatment in tuberculosis patients

Synne Jenum [1,11], Kristian Tonby[1,2,11], Corina S. Rueegg [3], Morten Rühwald[4,5], Max P. Kristiansen [6], Peter Bang [6], Inge Christoffer Olsen [7], Kjersti Sellæg [1], Kjerstin Røstad[1], Tehmina Mustafa [8,9], Kjetil Taskén [2,10], Dag Kvale[1,2], Rasmus Mortensen [4] & Anne Ma Dyrhol-Riise [1,2 ✉]

Host-directed-therapy strategies are warranted to fight tuberculosis. Here we assess the safety and immunogenicity of adjunctive vaccination with the H56:IC31 candidate and cyclooxygenase-2-inhibitor treatment (etoricoxib) in pulmonary and extra-pulmonary tuberculosis patients in a randomized open-label phase I/II clinical trial (TBCOX2, NCT02503839). A total of 222 patients were screened, 51 enrolled and randomized; 13 in the etoricoxib-group, 14 in the H56:IC31-group, 12 in the etoricoxib+H56:IC31-group and 12 controls. Three Serious Adverse Events were reported in the etoricoxib-groups; two urticarial rash and one possible disease progression, no Serious Adverse Events were vaccine related. H56:IC31 induces robust expansion of antigen-specific T-cells analyzed by fluorospot and flow cytometry, and higher proportion of seroconversions. Etoricoxib reduced H56:IC31-induced T-cell responses. Here, we show the first clinical data that H56:IC31 vaccination is safe and immunogenic in tuberculosis patients, supporting further studies of H56:IC31 as a host-directed-therapy strategy. Although etoricoxib appears safe, our data do not support therapy with adjunctive cyclooxygenase-2-inhibitors.

[1] Department of Infectious diseases, Oslo University Hospital, P.O box 4952 Nydalen N-0424 Oslo, Norway. [2] Institute of Clinical Medicine, University of Oslo, P.O box 1171 Blindern N-0318 Oslo, Norway. [3] Oslo Centre for Biostatistics and Epidemiology, Oslo University Hospital, P.O box 4950 Nydalen N-0424 Oslo, Norway. [4] Center for Vaccine Research, Department of Infectious Disease Immunology, Statens Serum Institut Artillerivej 5, 2300 Copenhagen, Denmark. [5] Foundation of Innovative New Diagnostics (FIND), the global alliance for diagnostics, Chemin des Mines 9, 1201 Geneva, Switzerland. [6] Center for Vaccine Research, Vaccine Development, Statens Serum Institut, Artillerivej 5, 2300 Copenhagen, Denmark. [7] Department of Research Support for Clinical Trials,, Oslo University Hospital, P.O box 4950 Nydalen N-0424 Oslo, Norway. [8] Centre for International Health, Department of Global Public Health and Primary Care, University of Bergen, P.O box 7804, N-5020 Bergen, Norway. [9] Department of Thoracic Medicine, Haukeland University Hospital, P.O box 1400, N-5021 Bergen, Norway. [10] Department of Cancer Immunology, Institute for Cancer Research, Oslo University Hospital, Montebello 0310 Oslo, Norway. [11] These authors contributed equally: Synne Jenum, Kristian Tonby. ✉email: a.m.d.riise@medisin.uio.no

G lobally, tuberculosis (TB) caused by *Mycobacterium tuberculosis* (Mtb) is the largest killer among infectious diseases causing 1·4 million deaths in 2018[1]. Long treatment duration and frequent adverse drug reactions challenge treatment adherence in TB patients and make treatment failure and relapse a common problem. Clinical trials of shorter treatment duration in drug-sensitive (DS) TB have failed, and treatment regimens for the increasing multi-drug resistant (MDR) TB are considerably longer and more toxic[2]. New affordable and shorter treatment modalities applicable in resource-limited settings are therefore warranted.

Host-directed therapy (HDT) aiming at amplifying host immunity to enhance Mtb killing or containment, has gained increasing research interest as an avenue to improve cure rates. HDT might also reduce morbidity by modulating inflammation and lung-tissue destruction[3,4]. In this regard, leading stakeholders have suggested to repurpose broadly used drugs like non-steroidal anti-inflammatory drugs (NSAIDs), including cyclooxygenase inhibitors (COX-i), as adjunctive to standard TB treatment. Currently, there is conflicting preclinical evidence on the effect of COX-i treatment in experimental Mtb infection: Reduced bacterial burden and limited lung pathology were demonstrated after intravenous infection[5–7], whereas increased bacterial burden and impaired cellular immunity were observed after low-dose aerosol infection[7]. The fact that COX-i are already widely used to alleviate TB symptoms and side-effects during TB treatment, further emphasize the need for exploring the effect of COX-i on immune responses during human TB disease.

Therapeutic vaccination as HDT represents an appealing strategy for treatment shortening and enhanced immune-mediated Mtb control independent of drug resistance[8,9], likely contributing to improved treatment outcome and reduced risk of recurrence[9]. The H56:IC31 subunit vaccine[10] consisting of the antigens Ag85B secreted early in disease, ESAT-6 constitutively expressed and Rv2660c associated with Mtb latency, augment both cellular and humoral immune responses[11]. The proven efficacy of H56:IC31 in mice and non-human primate (NHPs) models makes H56:IC31 a highly relevant candidate in therapeutic vaccination trials[10,12–15]. Moreover, H56:IC31 is safe and immunogenic in both Mtb infected and uninfected individuals[16–18] with no reported safety concerns in adults vaccinated within one month after finalizing TB treatment (NCT02375698, unpublished). Still, safety and immunogenicity are unknown in patients during ongoing TB disease.

In this work, we show that H56:IC31 vaccination is safe and immunogenic in TB patients. COX-2i adjunctive to standard treatment was safe, but did not improve TB immunity and reduced H56:IC31-induced T-cell responses. Our data support further studies of H56:IC31 vaccine as a HDT strategy.

## Results

**Study participants**. A total of 222 patients were assessed for eligibility at both study sites from November 2015 to January 2019 (Fig. 1). All enrolled participants ($N = 51$) were randomized. The safety analysis set (SAS) included 47 patients until loss of follow-up or end of study, and consisted of 13 patients allocated to etoricoxib treatment for 140 days, 12 patients to H56:IC31 vaccination at day 84 and 140, 10 patients to 140 days of etoricoxib and H56:IC31 vaccination at day 84 and 140, and 12 controls receiving standard of care with TB treatment only. The mean follow-up time and details on discontinuations are given in Supplementary Information (p. 13).

Baseline characteristics of the study participants in the SAS are listed in Table 1. The median age was 27 years (range 18–64) and 53% were men. The majority of patients had pulmonary TB;

etoricoxib-group 11/13 (85%), H56:IC31-group 12/12 (100%), controls 9/12 (75%) and etoricoxib+H56:IC31-group 7/10 (70%). Judged by TB symptom score (two or more of cough, chest pain or night sweat), the controls (3/12, 25%) and the etoricoxib +H56:IC31-group (1/10, 10%) had milder disease compared to the etoricoxib-group (10/13, 77%) and H56:IC31-group (9/12, 75%). Otherwise no differences in demographic and clinical baseline characteristics were observed in the SAS population. The distributions of baseline characteristics between the intervention groups in the Full Analysis Set (FAS) population were comparable to the SAS population and are presented in Supplementary Information (p. 12).

Adherence to etoricoxib was satisfactory with 85% of participants in the SAS taking ≥80% of prescribed tablets within the etoricoxib-group and 100% within the etoricoxib+H56:IC31-group (Supplementary Information pp. 14–15). Etoricoxib was detectable in plasma 14 and 84 days after treatment initiation in all patients (Supplementary Information p 11). The H56:IC31-vaccine was administered once in 11/12 (92%) and twice in 10/12 (83%) participants in the H56:IC31-group (one participant withdrew before the first vaccination). In the etoricoxib +H56:IC31-group H56:IC31 was administered once in 8/10 (80%) and twice in 8/10 (80%) participants. Two participants in the etoricoxib+H56:IC31-group experienced serious adverse events (SAEs) and were therefore excluded before the first vaccination. Adherence within the FAS was comparable to SAS (Supplementary Information pp. 14–15).

**Safety analysis of etoricoxib and H56:IC31**. In the period day 0 to day 84 we assessed safety and tolerability of adjunctive etoricoxib (etoricoxib-group and etoricoxib + H56:IC31-group before vaccination) compared to standard TB treatment only (control + H56:IC31-group before vaccination) (Table 2 and Supplementary Information pp. 16–18). Three SAEs were reported in the etoricoxib-groups; urticarial rash, a well-known side-effect of COX-2i, was observed in two participants, and possible disease progression was observed in one participant. All SAEs lead to discontinuation of etoricoxib. There was one registered suspected unexpected serious adverse reaction (SUSAR) in the etoricoxib-group due to urticarial rash, which in retrospect should have been classified as an adverse event (AE). Two SAEs were reported in controls: cerebral vasculitis and hemoptysis that occurred in the same participant. In general, the most frequent AEs (≥4 occurrences) were fatigue, nausea, arthralgia, and headache. The numbers of participants experiencing at least one AE and the total numbers of AEs were evenly distributed between the groups. With regard to possible intervention-related AEs, the numbers of any reported AE were higher in the etoricoxib-groups (etoricoxib: 77%, etoricoxib+H56:IC31: 90%) compared to controls or the H56:IC31-group (no listed events).

In the second safety period, day 85 to day 154, we assessed safety and tolerability in all groups (Table 2 and Supplementary Information pp. 19–20). Notably, no SAEs or SUSARs were reported in this time interval. One patient (HIV negative) developed enlargement of a cervical gland interpreted as a possible immune reconstitution inflammatory syndrome (IRIS) reported as an AE and leading to discontinuation of the assigned second H56:IC31-vaccination. The numbers of participants experiencing at least one AE or possible intervention-related AE, were higher in the H56:IC31-group (82% and 55%) compared to the etoricoxib-group (64% and 36%), the etoricoxib+H56:IC31-group (63% and 38%), and controls (50% and NA). The most frequent AEs in the H56:IC31-group were arthralgia, fatigue, elevated hepatic enzyme, nausea, and abdominal pain. The number of patients experiencing the most frequent AEs was comparable between the study groups.

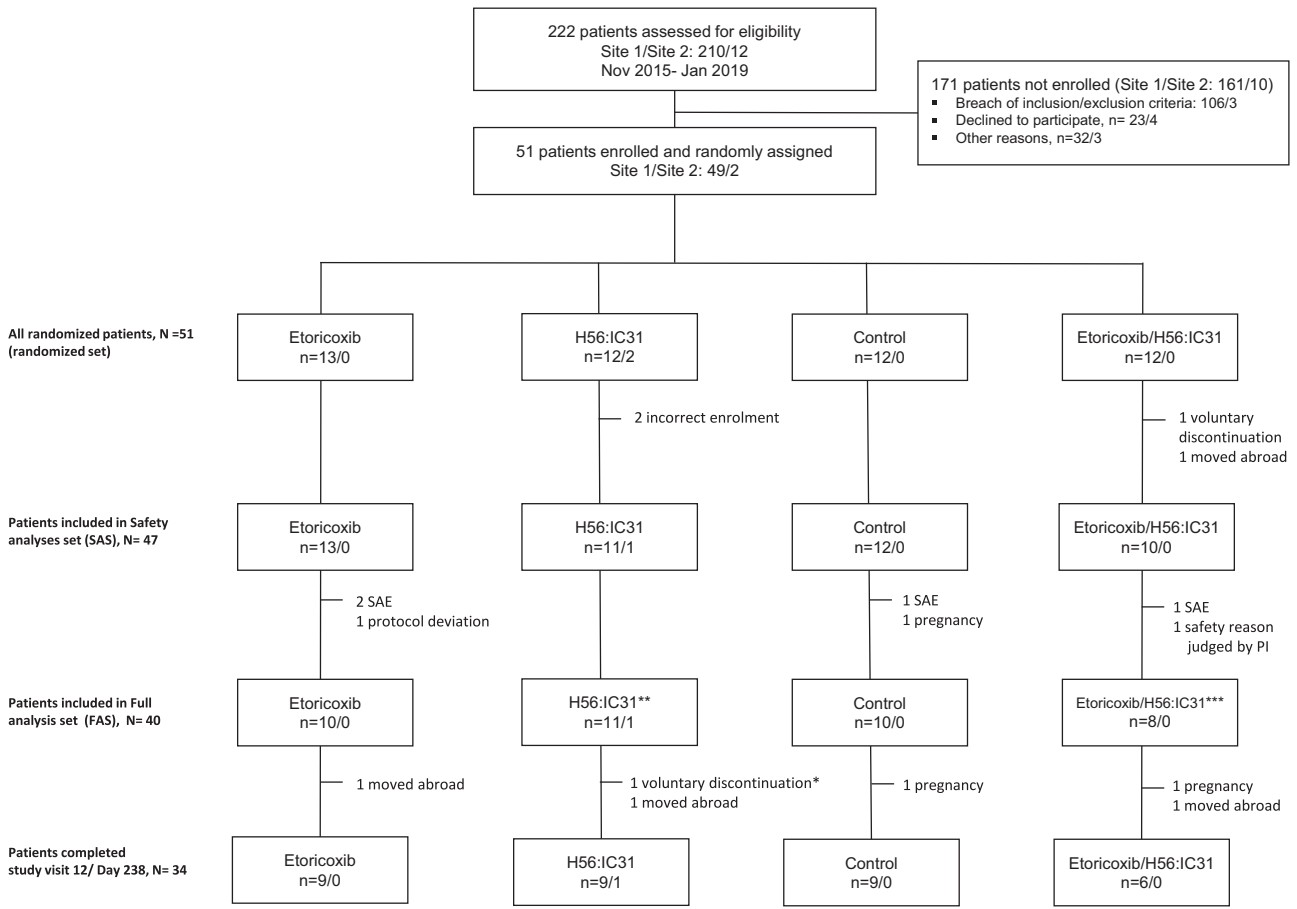

**Fig. 1 Study flow chart.** Participants were assessed for eligibility before enrollment and randomization to adjunctive (i) etoricoxib, (ii) H56:IC31, (iii) controls, or (iv) etoricoxib+H56:IC31. All randomized participants were included in the randomized set. Patients that received at least one dose of etoricoxib and/or one dose of H56:IC31 were defined as the safety analysis set (SAS). The secondary outcomes of immunogenicity were analyzed in the full analysis set (FAS) defined as participants within SAS with at least one valid immunogenicity measurement. Reasons for study discontinuation are denoted in the flowchart. *Participant sampled day 84 but withdrew prior to the first administration of H56:IC31. **10 patients received two doses of H56:IC31. ***8 patients received two doses of H56:IC31.

In the last safety period, day 155 to day 238 (end of study), we assessed long-term AEs occurring across all groups after ending all interventions (Table 2 and Supplementary Information pp. 21–22). One SAE, duodenitis (verified by gastroscopy) was observed in the etoricoxib-group. No SUSARs were recorded. The total numbers of AEs were higher both in the etoricoxib (70%) and the H56:IC31 (73%) groups compared to controls (44%) and the etoricoxib+H56:IC31-group (38%).

**Immunoregulatory effects of etoricoxib**. We first evaluated the effects of etoricoxib administered during the first 84 days of standard TB treatment on Mtb-specific immunity comparing the etoricoxib and etoricoxib+H56:IC31 groups to controls and the H56:IC31-group (before vaccination). There was no significant difference in the changes in median Cytokine$^+$ T-cell responses obtained by the fluorescence IFNγ/IL-2 immuno-spot (Fluorospot) assay (Fig. 2a), Cytokine$^+$ CD4 T-cell responses from whole blood intracellular cytokine staining (WB-ICS) (Fig. 2b, c), or other primary ranked read-outs (Table 3, Supplementary Fig. 2, p. 24). However, a trend of decreased Cytokine$^+$ CD4 T-cell responses to purified protein derivative (PPD) was observed in the etoricoxib groups (WB-ICS) (Fig. 2b and Table 3).

**Immunogenicity of the H56:IC31 vaccine**. When evaluating the effects of H56:IC31 administered at two doses (day 84 and day

140) during standard TB treatment, we observed a significant increase in primary ranked T-cell responses measured by Fluorospot in the H56:IC31-group compared to controls at day 154 (Table 3). This was evident for both Cytokine$^+$ T-cell responses to the H56 fusion protein (Fig. 3a, c) and to the individual vaccine peptides Ag85B and ESAT-6 summed (Fig. 3b). The relative difference was most prominent for Cytokine$^+$ T-cell responses to Ag85B in the H56:IC31-group, and these responses remained increased at day 238 (Fig. 3d and Supplementary Fig. 3, p. 25). The same pattern was observed for IFNγ$^+$ T-cell responses to H56 (Supplementary Fig. 4, p. 28). No apparent differences in CFP-10 specific Cytokine$^+$ T-cells responses between groups were observed indicating no additional H56:IC31 effect on Mtb-induced immunity since CFP-10 is not included in the vaccine (Supplementary Fig. 3, p. 26).

WB-ICS analysis showed the same pattern of immune responses in the H56:IC31-group as observed in the Fluorospot assay, although the differences in total Cytokine$^+$ CD4 T-cell responses to H56 peptides (sum Ag85B, ESAT-6, and Rv2660c) were not statistically significant (Table 3, Fig. 3e).

T-cells expressing IL-2 in combination with TNFα and/or IFNγ are considered important for long-term protective immunity and thus of particular interest for TB vaccine development[19,20]. Therefore, the striking findings in the Fluorospot data of higher median frequencies of IL-2$^+$ and IFNγ$^+$IL-2$^+$ producing T-cells in the H56:IC31-group compared to controls, especially after

**Table 1 Baseline characteristics by study arm for the safety analysis set (SAS), _N_ = 47.**

| | Etoricoxib (_N_ = 13) | H56:IC31 (_N_ = 12) | Controls (_N_ = 12) | Etoricoxib + H56:IC31 (_N_ = 10) |
|---|---|---|---|---|
| _Demography_ | | | | |
| Gender (Male) | 7 (54%) | 8 (67%) | 5 (42%) | 5 (50%) |
| Age (years), Median (Q1, Q3) | 34 (25, 36) | 24 (21, 27) | 28 (23, 33) | 28 (25, 34) |
| Ethnicity | | | | |
| Asian | 4 (31%) | 2 (17%) | 5 (42%) | 5 (50%) |
| Black | 5 (38%) | 7 (58%) | 6 (50%) | 3 (30%) |
| Caucasian | 2 (15%) | 3 (25%) | 1 (8%) | 2 (20%) |
| Other | 2 (15%) | 0 (0%) | 0 (0%) | 0 (0%) |
| _TB-related Medical history_ | | | | |
| BCG scar | | | | |
| Yes | 5 (38%) | 5 (42%) | 6 (50%) | 7 (70%) |
| No | 7 (54%) | 6 (50%) | 6 (50%) | 1 (10%) |
| Unknown | 1 (8%) | 1 (8%) | 0 (0%) | 2 (20%) |
| Previous TB treatment | 0 (0%) | 1 (8%) | 2 (17%) | 2 (20%) |
| _Co-morbidity and Risk factors_ | | | | |
| Allergy | 3 (23%) | 1 (8%) | 3 (25%) | 4 (40%) |
| Previous gastric ulcer/gastroenteric bleeding | 1 (8%) | 1 (8%) | 1 (8%) | 0 (0%) |
| Liver disease/failure | 1 (8%) | 0 (0%) | 1 (8%) | 0 (0%) |
| Pulmonary disease | 1 (8%) | 0 (0%) | 0 (0%) | 0 (0%) |
| Other medical conditions* | 3 (23%) | 3 (25%) | 3 (25%) | 4 (40%) |
| Smoking status | 3 (23%) | 0 (0%) | 0 (0%) | 1 (10%) |
| Alcohol use >3 units/week | 3 (23%) | 1 (8%) | 2 (17%) | 4 (40%) |
| Drug abuse | 0 (0%) | 1 (8%) | 0 (0%) | 0 (0%) |
| _Clinical classification and radiology_ | | | | |
| Clinical classification | | | | |
| Pulmonary | 9 (69%) | 10 (83%) | 8 (67%) | 6 (60%) |
| Pulmonary/extrapulmonary | 2 (15%) | 2 (17%) | 1 (8%) | 1 (10%) |
| Extrapulmonary | 2 (15%) | 0 (0%) | 3 (25%) | 3 (30%) |
| Lymph glands | 2 | 1 | 4 | 4 |
| Bones, joint and soft tissue | 0 | 0 | 0 | 1 |
| Other | 2 | 1 | 0 | 0 |
| Radiology (pulmonary cases) | | | | |
| Infiltrate | 8 | 11 | 5 | 4 |
| Cavity | 5 | 3 | 2 | 0 |
| Pleural fluid | 1 | 1 | 0 | 0 |
| Hilary/mediastinal changes | 0 | 2 | 0 | 0 |
| Other | 1 | 1 | 1 | 0 |
| _Clinical features_ | | | | |
| Symptoms (cough, chest pain, night sweat) | | | | |
| No symptoms | 2 (15%) | 2 (17%) | 8 (67%) | 6 (60%) |
| 1 symptom | 1 (8%) | 1 (8%) | 1 (8%) | 3 (30%) |
| ≥2 symptoms | 10 (77%) | 9 (75%) | 3 (25%) | 1 (10%) |
| BMI** (kg/m2) Median (Q1, Q3) | 21 (19, 22) | 21 (19, 22) | 22 (19, 24) | 22 (18, 26) |
| Body temperature (°C) Median (Q1, Q3) | 36.6 (36.5, 37.0) | 36.3 (36.1, 36.8) | 36.6 (36.4, 37.0) | 36.5 (36.2, 36.7) |
| _Laboratory data_ | | | | |
| Haemoglobin (g/dl) | | | | |
| Median (Q1, Q3) | 13.0 (12.2, 14.9) | 13.3 (12.3, 14.9) | 14.2 (13.4, 15.1) | 14.0 (13.3, 15.0) |
| N (% Non-missing) | 13 (100%) | 11 (92%) | 11 (92%) | 9 (90%) |
| Erythrocyte Sedimentation rate (mm) | | | | |
| Median (Q1, Q3) | 36.0 (12.0, 79.5) | 20.0 (11.0, 52.0) | 20.0 (5.0, 57.0) | 26.0 (15.0, 33.0) |
| N (% Non-missing) | 12 (92%) | 10 (83%) | 11 (92%) | 9 (90%) |
| C-reactive protein (mg/L) | | | | |
| Median (Q1, Q3) | 19.8 (3.0, 52.2) | 7.2 (0.7, 36.5) | 2.8 (0.0, 38.7) | 2.8 (0.9, 4.3) |
| N (% Non-missing) | 13 (100%) | 11 (92%) | 11 (92%) | 9 (90%) |
| QuantiFERON-TB | | | | |
| Negative | 1 (8%) | 0 (0%) | 2 (18%) | 0 (0%) |
| Positive | 11 (85%) | 10 (91%) | 8 (73%) | 10 (100%) |
| Indeterminate | 1 (8%) | 1 (9%) | 1 (9%) | 0 (0%) |
| Median IU/ml (Q1, Q3) | 5.0 (3.7, 7.2) | 4.8 (2.0, 8.0) | 6.4 (3.9, 11.3) | 8.0 (2.4, 13.9) |
| N (% Non-missing) | 11 (85%) | 10 (83%) | 8 (67%) | 10 (100%) |

Unless indicated in table, no values were missing.
*No participants had diabetes. **BMI: body mass index.

**Table 2 Adverse events (AE) by study group.**

| | Etoricoxib $N = 13$ (%) | H56:IC31 $N = 12$ (%) | Control $N = 12$ (%) | Etoricoxib + H56:IC31 $N = 10$ (%) |
|---|---|---|---|---|
| *Day 0 to day 84* | | | | |
| Participants with at least one AE | 11 (85%) | 11 (92%) | 11 (92%) | 9 (90%) |
| Number of AEs | 58 | 45 | 54 | 52 |
| Number of SAEs | 2 | 0 | 2 | 1 |
| Number of SUSARs | 1* | | | |
| *Relationship to study intervention* | | | | |
| Participants with at least one intervention-related AE | 10 (77%) | 0 | 0 | 9 (90%) |
| Participants with at least one intervention-related SAE | 2 (15%) | | | 1 (10%) |
| Number of AE leading to study drug discontinuation | 2 | | 1 | 7 |
| *Most frequent AE (4 or more occurrences) by patient ([no. of events] no. of patients (% of patients))* | | | | |
| Fatigue | [9] 8 (62%) | [4] 4 (33%) | [6] 6 (50%) | [3] 3 (30%) |
| Nausea | [7] 6 (46%) | [4] 4 (33%) | [3] 3 (25%) | [6] 5 (50%) |
| Arthralgia | [3] 3 (23%) | [3] 3 (25%) | [6] 6 (50%) | [2] 2 (20%) |
| Headache | [6] 6 (46%) | [5] 5 (42%) | [2] 2 (17%) | [2] 1 (10%) |
| Hepatic enzyme increased | [8] 5 (39%) | [3] 3 (25%) | [3] 2 (17%) | [4] 3 (30%) |
| Myalgia | [3] 3 (23%) | [4] 2 (17%) | [2] 2 (17%) | [2] 2 (20%) |
| Pruritus | [2] 2 (15%) | [1] 1 (8%) | [2] 2 (17%) | [4] 3 (30%) |
| Rash | [1] 1 (8%) | [1] 1 (8%) | [3] 3 (25%) | [1] 1 (10%) |
| Dizziness | [3] 3 (23%) | | [1] 1 (8%) | [2] 2 (20%) |
| Dyspepsia | [1] 1 (8%) | [3] 2 (17%) | [2] 2 (17%) | |
| Decreased appetite | | | [4] 4 (33%) | [1] 1 (10%) |
| Cough | | [1] 1 (8%) | [2] 1 (8%) | [2] 2 (20%) |
| Night sweats | [1] 1 (8%) | | [2] 2 (17%) | [1] 1 (10%) |
| Chest pain | [1] 1 (8%) | [3] 3 (25%) | | |
| Abdominal pain upper | [1] 1 (8%) | [1] 1 (8%) | [2] 2 (17%) | |
| *Day 85 to day 154* | *N = 11 (%)* | *N = 11 (%)* | *N = 10 (%)* | *N = 8 (%)* |
| Participants with at least one AE | 7 (64%) | 9 (82%) | 5 (50%) | 5 (63%) |
| Number of AEs | 20 | 24 | 15 | 15 |
| Number of SAEs | 0 | 0 | 0 | 0 |
| Number of SUSARs | 0 | 0 | 0 | 0 |
| *Relationship to study intervention* | | | | |
| Participants with at least one intervention-related AE | 4 (36%) | 6 (55%) | | 3 (38%) |
| Participants with at least one intervention-related SAE | | | | |
| Number of AE leading to study drug discontinuation* | 0 | 1 (9%) | | 0 |
| *Most frequent AE (4 or more occurrences) by patient ([no. of events] no. of patients (% of patients))* | | | | |
| Arthralgia | [1] 1 (9%) | [4] 4 (36%) | | [1] 1 (13%) |
| Fatigue | [2] 2 (18%) | [3] 3 (27%) | | [1] 1 (13%) |
| Hepatic enzyme increased | [1] 1 (9%) | [2] 2 (18%) | [1] 1 (10%) | |
| Nausea | [1] 1 (9%) | [1] 1 (9%) | [2] 1 (10%) | [1] 1 (13%) |
| Abdominal pain | [1] 1 (9%) | [1] 1 (9%) | [1] 1 (10%) | [1] 1 (13%) |
| *Local solicited adverse events* | | | | |
| Local reaction | | [1] 1 (9%) | | [1] 1 (13%) |
| Vaccination site induration | | | | [1] 1 (13%) |
| Vaccination site pain | | [1] 1 (9%) | | [2] 2 (25%) |
| *Day 155 to day 238* | *N = 10 (%)* | *N = 11 (%)* | *N = 9 (%)* | *N = 8 (%)* |
| Participants with at least one AE | 7 (70%) | 8 (73%) | 4 (44%) | 3 (38%) |
| Number of AEs | 12 | 14 | 7 | 8 |
| Number of SAEs | 1 | 0 | 0 | 0 |
| Number of SUSARs | 0 | 0 | 0 | 0 |
| Participants with at least one intervention-related AE | 3 (30%) | 3 (27%) | | 2 (25%) |
| Participants with at least one intervention-related SAE | 1 (10%) | 0 | 0 | 0 |
| *Most frequent AE (4 or more occurrences) by patient ([no. of events] no. of patients (% of patients))* | | | | |
| None | | | | |

AEs across all study groups during three time periods (day 0–84, day 85–154, day 155–238).
*1 patient registered with study drug interruption, but since not receiving second vaccine dose, the AE was re-categorized to "AE leading to study drug discontinuation".

Ag85B stimulation (Supplementary Fig. 4b, p. 29), motivated us to proceed with post-hoc analyses of the T-cell responses in the H56:IC31-group (Table 3, Fig. 3f). WB-ICS data revealed an overall expansion of polyfunctional CD4 T-cell subsets after vaccination within the H56:IC31-group compared to controls for the following subsets: (i) Triple-producing IFNγ+IL2+TNFα+ CD4 T-cells, with significant difference observed for responses to Ag85B; (ii) Duo-producing IFNγ-IL2+TNFα+ CD4 T-cells, (iii) Duo-producing IFNγ+IL2+TNFα− CD4 T-cells, with significant difference observed for responses to Ag85B and (iv) Duo-producing IFNγ+IL2-TNFα+ CD4 T-cells, with significant differences observed for responses to Ag85B and all peptides summed. Various cytokine responses to the individual peptides are shown in Supplementary Fig. 7, pp. 32–42.

Despite induction of robust T-cell responses, H56:IC31 did not evoke a significant increase in anti-H56 IgG at any time point

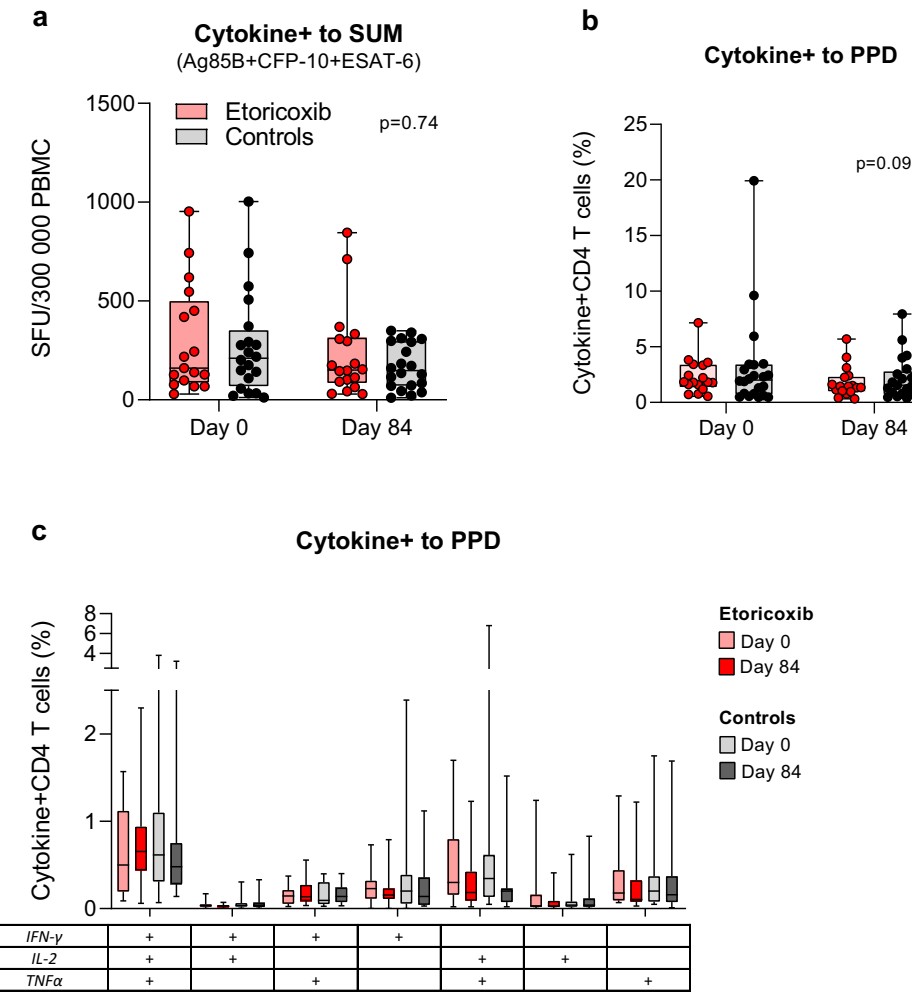

**Fig. 2 Immunoregulatory effects of etoricoxib.** Effect of etoricoxib administered during the first 84 days of standard TB treatment on Mtb-specific immunity (Hypothesis 1, Table 3). **a** Cytokine+ T-cells in response to Ag85, ESAT-6 and CFP10 summed, in the etoricoxib group (red) and controls (gray) at day 0 (n: 17/20) and day 84 (n: 18/20). Data from Fluorospot analysis of thawed PBMCs isolated at day 0 and day 84 and stimulated with 15mer overlapping individual peptide pools (>80–85% purity) at 2 μg/ml for 17 h in the presence of anti-CD28 (0.1 mg/ml) are shown. Cytokine+ was defined as the sum of IFNγ and IL-2 responses (total IFNγ plus total IL-2 minus duo IFNγ/IL-2). **b** Cytokine+ CD4+ T-cells in response to purified protein derivate (PPD) in the etoricoxib group (red) and controls (gray) at day 0 (n: 17/22) and day 84 (n: 17/21). Data from flow cytometry analysis with intracellular staining (ICS) of 1 ml peripheral whole blood (WB) incubated for 12 h (Brefeldin A added after 7 h) with PPD (10 μg/ml) in the presence of anti-CD28 and anti-CD49d (0.1 mg/ml) within 75 min of sampling at day 0 and day 84 are shown. Cytokine+ was defined as the sum of IFNγ, IL-2, and TNFα responses (total IFNγ plus duo IL-2/TNFα plus single IL-2 plus single TNFα). **c** Boolean gating was used to create CD4 T-cell subpopulations defined by their co-production of cytokines in response to stimulation with PPD in the etoricoxib group (red) and controls (gray) at day 0 (n: 17/22) and day 84 (n: 17/22). Boxplots are shown with median, interquartile ranges (IQR), and minimum/maximum values. Line plots are shown with IQR. Median regression with treatment as only independent explanatory factor was applied for immunogenicity read-outs pre-assigned a hierarchical order to compensate for lack of multiple testing adjustments. Effect estimates of primary and secondary priorities and 95% confidence intervals based on 1000 bootstrap replications and corresponding two-sided p-values are depicted in Table 3 and denoted in (**a**, **b**). Adjustments for multiple testing were not performed but a hierarchical order of immunogenicity outcomes defined a priori (Supplementary Information pp. 6–7) was judged adequate to assign importance/interpret the results of the respective statistical tests.

during follow-up in the H56:IC31 groups compared to controls (Fig. 3g). Still, there was an increase in titers from the first vaccination at day 84 to day 154, and titers were sustained above pre-vaccine levels at day 238. Serum conversion (>2 fold increase in anti-H56 IgG) occurred in 75% of vaccinated subjects compared to 60% in the etoricoxib-group and 50% in controls (Table 4).

**Adjunctive effects of etoricoxib on H56:IC31 immunogenicity.** Finally, we evaluated the hypothesized adjunctive effect of etor-icoxib on H56:IC31-induced immune responses given during standard TB treatment. Contradictory to our initial hypothesis,

but in line with the tendencies observed when comparing the etoricoxib-group to controls, etoricoxib did not increase H56:IC31-induced T-cell responses in the interval from day 84 to day 154 (Fig. 4). Instead, there was a significant decrease in H56:IC31 immunogenicity in the etoricoxib+H56:IC31-group compared to participants receiving H56:IC31 alone measured by both Fluorospot total Cytokine+ T-cell (Fig. 4a, c) and IFNγ+ T-cell responses after H56 stimulation (Table 3, Supplementary Fig. 5, p. 30). The same pattern was found for Cytokine+ T-cell responses to Ag85B and ESAT-6 summed (Table 3 and Fig. 4b) as well as responses to the individual peptides (Supplementary Fig. 6, p. 31). Evaluation by WB-ICS did not reveal significant

**Table 3 Outcomes of immunogenicity readouts according to hypothesis 1-3.**

| | | Method | Readouts | Estimated median difference with 95% CI | p-values |
|---|---|---|---|---|---|
| PRIMARY PRIORITY | *Hypothesis 1* Day 0–84 etoricoxib vs controls | FLUOROSPOT (FS) | Cytokine+ to SUM Ag85B+ESAT6+CFP10 | −19.3 (−136 to 97.7) | 0.74 |
| | | | IFNγ to SUM Ag85B+ESAT6+CFP10 | −28.3 (−128.9 to 72.2) | 0.57 |
| | | Whole blood ICS (WB-ICS) | Cytokine+ to PPD | −0.47 (−1.02 to 0.071) | 0.09 |
| | | | Cytokine+ to SUM Ag85B+ESAT6+Rv2660c | −0.18 (−0.46 to 0.10) | 0.21 |
| | *Hypothesis 2* Day 84–154 H56:IC31 vs controls | FLUOROSPOT (FS) | Cytokine+ to H56 | 149.7 (22.0 to 277.3) | **0.02** |
| | | | IFNγ to H56 | 93.8 (25.2 to 162.4) | **0.01** |
| | | Whole blood ICS (WB-ICS) | Cytokine+ to SUM Ag85B+ESAT6 | 214.3 (11.77 to 416.9) | **0.04** |
| | | | Cytokine+ to SUM Ag85B+ESAT6+Rv2660c | 0.33 (−0.78 to 1.44) | 0.54 |
| | | ELISA serology | IgG antibody to H56 | 1.15 (−3.89 to 6.1) | 0.64 |
| | *Hypothesis 3* Day 84–154 etoricoxib+H56:IC31 vs H56:IC31 | FLUOROSPOT (FS) | Cytokine+ to H56 | −176.3 (−352.0 to −0.6) | 0.05 |
| | | | IFNγ to H56 | −109.2 (−215.7 to −2.7) | 0.05 |
| | | Whole blood ICS (WB-ICS) | Cytokine+ to SUM Ag85B+ESAT6 | −252.7 (−495.6 to −9.7) | **0.04** |
| | | | Cytokine+ to SUM Ag85B+ESAT6+Rv2660c | −0.37 (−1.57 to 0.84) | 0.53 |
| | | ELISA serology | IgG antibody to H56 | −0.86 (−20.8 to 19.1) | 0.93 |
| SECONDARY PRIORITY | *Hypothesis 2* Day 84–154 H56:IC31 vs controls | Whole blood ICS (WB-ICS) | IFNγ+ IL2+ TNFα+ CD4 to Ag85B | 0.04 (0.00 to 0.090) | **0.05** |
| | | | IFNγ+ IL2+ TNFα+ CD4 to ESAT6 | 0.01 (−0.08 to 0.10) | 0.79 |
| | | | IFNγ+ IL2+ TNFα+ CD4 to Rv2660c | 0.00 (−0.01 to 0.01) | 0.79 |
| | | | IFNγ+ IL2+ TNFα+ CD4 to SUM Ag85B+ESAT6+Rv2660c | 0.09 (−0.04 to 0.21) | 0.16 |
| | | | IFNγ- IL2+ TNFα+ CD4 to Ag85 | 0.01 (−0.01 to 0.04) | 0.30 |
| | | | IFNγ- IL2+ TNFα+ CD4 to ESAT6 | 0.02 (−0.02 to 0.05) | 0.32 |
| | | | IFNγ- IL2+ TNFα+ CD4 to Rv2660c | 0.00 (−0.00 to 0.00) | 1.00 |
| | | | IFNγ- IL2+ TNFα+ CD4 to SUM Ag85B+ESAT6+Rv2660c | 0.03 (−0.02 to 0.08) | 0.29 |
| | | | IFNγ+ IL2+ TNFα− CD4 to Ag85B | 0.02 (0.00 to 0.03) | **0.03** |
| | | | IFNγ+ IL2+ TNFα− CD4 to ESAT6 | 0.01 (−0.00 to 0.02) | 0.12 |
| | | | IFNγ+ IL2+ TNFα− CD4 to Rv2660c | 0.00 (−0.00 to 0.01) | 0.58 |
| | | | IFNγ+ IL2+ TNFα− CD4 to SUM Ag85B+ESAT6+Rv2660c | 0.05 (−0.00 to 0.09) | 0.05 |
| | | | IFNγ+ IL2- TNFα+ CD4 to Ag85 | 0.05 (0.01 to 0.08) | **0.01** |
| | | | IFNγ+ IL2- TNFα+ CD4 to ESAT6 | 0.02 (−0.00 to 0.04) | 0.08 |
| | | | IFNγ+ IL2- TNFα+ CD4 to Rv2660c | 0.01 (−0.00 to 0.03) | 0.13 |
| | | | IFNγ+ IL2- TNFα+ CD4 to SUM Ag85B+ESAT6+Rv2660c | 0.10 (0.02 to 0.17) | **0.02** |

Table 3 Outcomes of immunogenicity analyses readouts for hypothesis 1–3 according to pre-assigned hierarchical order in the statistical plan to compensate for lack of multiple testing and facilitate interpretation. For Hypothesis 1 etoricoxib consists of both the etoricoxib group and the etoricoxib+H56:IC31 group (before vaccination) and controls consists of both the control group and the H56:IC31 group (before vaccination). Upper part of table shows primary priority outcomes. Lower part of table shows posthoc analysis of readouts defined as secondary priority outcomes. The treatment differences were calculated as estimated difference in median response with 95% confidence intervals based on 1000 bootstrap replications and corresponding two-sided p-value. The significant p-values are in bold.

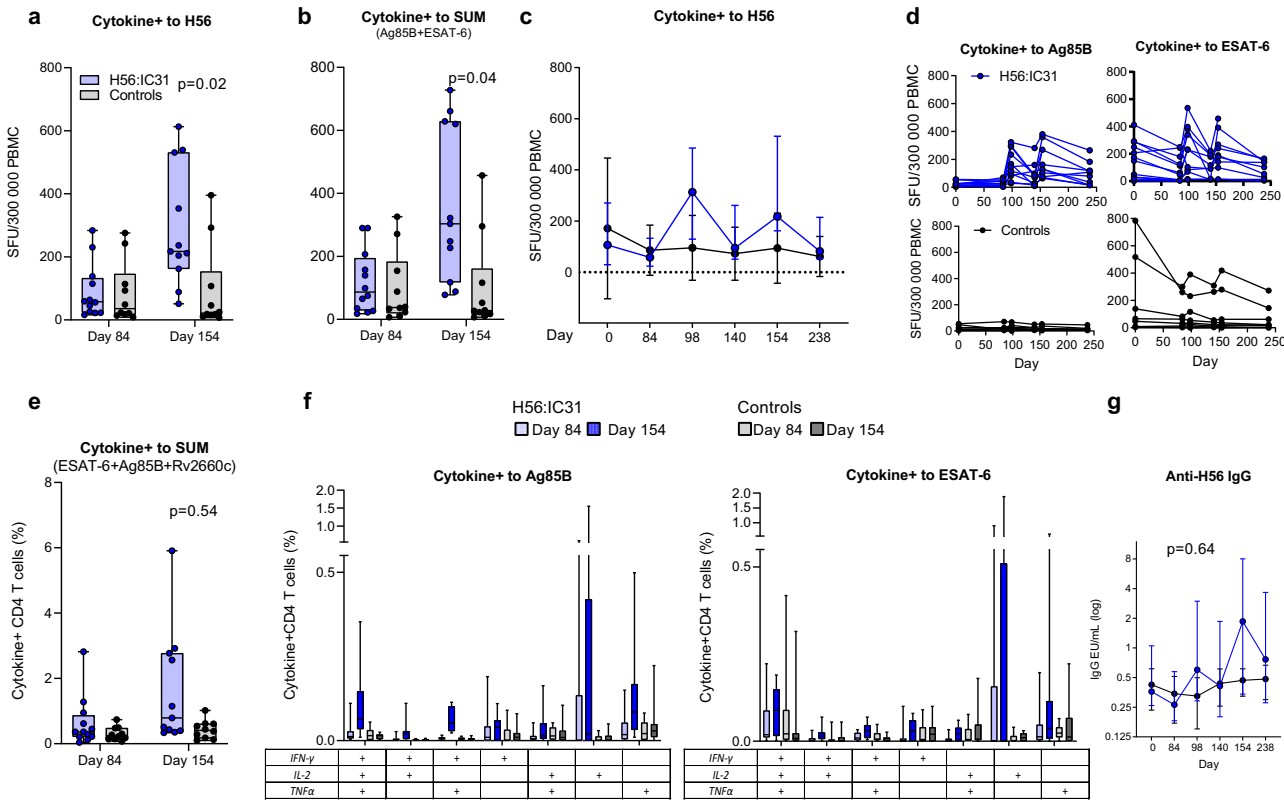

**Fig. 3 Immunogenicity of the H56:IC31 vaccine.** Effect of H56:IC31 administered at two doses (day 84 and day 140) on vaccine-induced immunity in the H56:IC31 group (blue; $n = 12$) and controls (gray, $n = 12$) (Hypothesis 2, Table 3). **a**, **b**, **c** and **d** depicts data from Fluorospot analysis of thawed PBMCs, isolated at day 84 and 154 ($n$ for H56:IC31/$n$ for controls) and stimulated with H56 fusion protein, Ag85B and ESAT-6* in the presence of anti-CD28 (0.1 mg/ml) for 17 h. Cytokine+ was defined as the sum of IFNγ and IL-2 responses (total IFNγ plus total IL-2 minus duo IFNγ/IL-2). **a** Cytokine+ T-cells in response to a H56 fusion protein at day 84 ($n$: 12/10) and day 154 ($n$: 11/10), **b** Ag85B and ESAT-6 summed at day 84 ($n$: 12/10) and day 154 ($n$: 11/10). **c** Longitudinal Cytokine+ T-cell responses to H56 from day 0 to day 238. **d** Individual trajectories of Cytokine+ T-cell responses to Ag85B and ESAT-6 stratified by intervention. **e**, **f** Depicts data from flow cytometry analysis with intracellular staining (ICS) of 1 ml peripheral whole blood (WB) incubated for 12 h (Brefeldin A added after 7 h) with Ag85B, ESAT-6, and Rv2660c* in the presence of anti-CD28 and anti-CD49d (0.1 mg/ml) within 75 min of sampling at day 84 and day 154. Cytokine+ was defined as the sum of IFNγ, IL-2 and TNFα responses (total IFNγ plus duo IL-2/TNFα plus single IL-2 plus single TNFα). **e** Cytokine+ CD4 T-cells in response to Ag85B, ESAT-6, and Rv2660c summed at day 84 ($n$: 12/10) and day 154 ($n$: 11/10). **f** Boolean gating was used to create CD4 T-cell subpopulations defined by their co-production of cytokines in response to Ag85B and ESAT-6 at day 84 ($n$: 12/10) and day 154 ($n$: 11/10). **g** Longitudinal log transformed anti-H56 IgG serum levels from day 0 ($n$: 12/10) to day 238 ($n$: 10/6) analyzed by in-house ELISA. *Stimulants in Fluorospot and ICS as follows: the H56 fusion protein at 5 µg/ml, the Ag85B, ESAT-6, and Rv2660c (15mer overlapping individual peptide pools >80–85% purity at 2 µg/ml). Boxplots are shown with median, interquartile ranges (IQR) and minimum/maximum values. Line plots are shown with IQR. Median regression with treatment as only independent explanatory factor was applied for immunogenicity read-outs pre-assigned a hierarchical order to compensate for lack of multiple testing adjustments. Effect estimates of primary priority and post-hoc analyses of secondary priority, 95% confidence intervals based on 1000 bootstrap replications and corresponding two-sided $p$-values depicted in Table 3 and in (**a**, **b**, **e**, **g**).

**Table 4 Serum conversion following H56:IC31 vaccination.**

|  | Proportion of converters N (%) |
|---|---|
| Etoricoxib ($N = 10$) | 6 (60%) |
| H56:IC31 vaccine ($N = 12$) | 9 (75%) |
| Control ($N = 10$) | 5 (50%) |
| Etoricoxib + H56:IC31 ($N = 8$) | 6 (75%) |

Serum conversions following H56:IC31 vaccination in participants stratified by intervention group. Conversion was defined by >2 fold increase in anti-H56 IgG level from day 84 to any time point between day 98 and 238.

differences in Cytokine+ CD4 T-cell responses to summed H56 peptides (Table 3, Fig. 4e). However, the positive effects observed for CD4 T-cell subpopulations in the H56:IC31-group were attenuated in the combined etoricoxib+H56:IC31-group (Supplementary Fig. 7 pp. 32–42).

Finally, etoricoxib adjunctive to H56:IC31-vaccination did not increase anti-H56 IgG levels significantly above the levels induced by H56:IC31 alone (Fig. 4g). In contrast to T-cell responses, anti-H56:C31 IgG levels were not reduced by adjunctive etoricoxib, and the proportions of seroconverters were 75% in both groups (Table 4).

## Discussion

The present trial assessed the safety and immune-modulating capacity of three possible strategies for HDT in patients with DS-TB. For the first time in humans, we have tested the subunit H56:IC31 TB vaccine candidate as a therapeutic vaccine during TB disease. H56:IC31 is already proven safe and immunogenic in BCG-vaccinated healthy adults with and without Mtb infection[16–18] and in successfully treated adult TB patients (NCT02375698, unpublished). With the present study we report no major safety concerns with two injections of H56:IC31

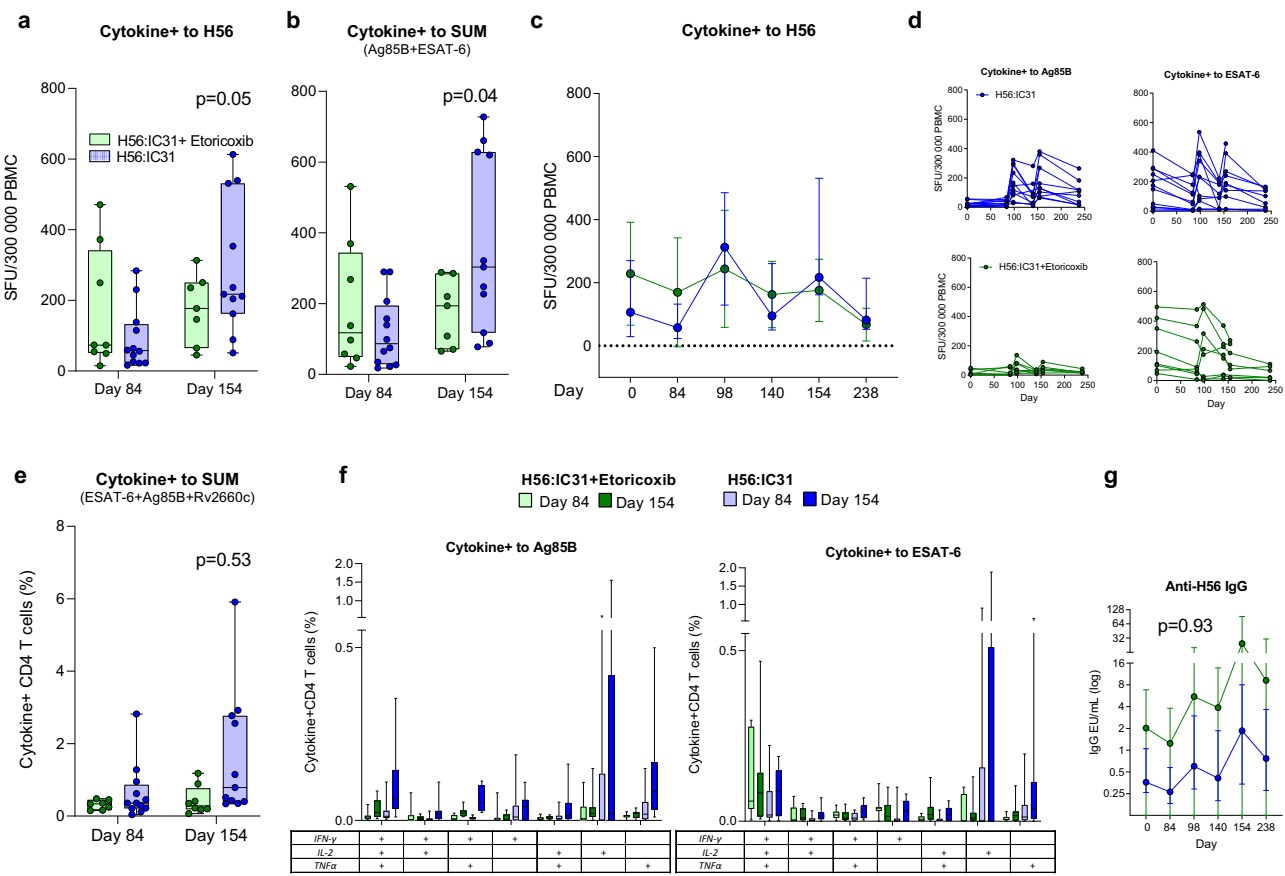

**Fig. 4 Adjunctive effects of etoricoxib on H56:IC31 immunogenicity.** Effect of etoricoxib given during the first 140 days on vaccine-induced immunity after two doses of H56:IC31 in the etoricoxib+H56:IC31-group (green, n = 8) and H56:IC31-group (blue, n = 12) (days 84 and 140) (Hypothesis 3, Table 3). **a**, **b**, **c**, and **d** Depicts data from Fluorospot analysis of thawed PBMCs isolated at days 84 and 154 (n for etoricoxib+H56:IC31/n for H56:IC31) and stimulated with H56 fusion protein, Ag85B, and ESAT-6* in the presence of anti-CD28 (0.1 mg/ml) for 17 h. Cytokine+ was defined as the sum of IFNγ and IL-2 responses (total IFNγ plus total IL-2 minus duo IFNγ/IL-2). Cytokine+ T-cells in response to **a** H56 fusion protein at day 84 (n: 8/12) and day 154 (n: 7/11), **b** Ag85B and ESAT-6 summed at day 84 (n: 8/12) and day 154 (n: 7/11), **c** Longitudinal Cytokine+ T-cells in response to H56 from day 0 to 238. **d** Individual trajectories showing Cytokine+ T-cell in response to Ag85B and ESAT-6 stratified by intervention. **e**, **f** Depicts data from flow cytometry analysis with intracellular staining (ICS) of 1 ml peripheral whole blood (WB) incubated for 12 h (Brefeldin A added after 7 h) with Ag85B, ESAT-6, and Rv2660c in the presence of anti-CD28 and anti-CD49d (0.1 mg/ml) within 75 min of sampling at days 84 and 154. Cytokine+ was defined as the sum of IFNγ, IL-2, and TNFα responses (total IFNγ plus duo IL-2/TNFα plus single IL-2 plus single TNFα). **e** Cytokine+ CD4 T-cells in response to Ag85B, ESAT-6, and Rv2660c summed at day 84 (n: 7/12) and day 154 (n: 8/11). **f** Boolean gating was used to create CD4 T-cell subpopulations defined by their co-production of cytokines in response to Ag85B and ESAT-6 at days 84 (n: 7/12) and 154 (n: 8/11). **g** Longitudinal log transformed anti-H56 IgG serum levels from day 0 (n: 8/12) to 238 (n: 6/10) analyzed by in-house ELISA. *Stimulants in Fluorospot and ICS as follows: the H56 fusion protein at 5 µg/ml, the Ag85B, ESAT-6, and Rv2660c (15mer overlapping individual peptide pools >80–85% purity at 2 µg/ml). Boxplots are shown with median, interquartile ranges (IQR), and minimum/maximum values. Line plots are shown with IQR. Median regression with treatment as only independent explanatory factor was applied for immunogenicity read-outs pre-assigned a hierarchical order to compensate for lack of multiple testing adjustments. Effect estimates of primary priority and 95% confidence intervals based on 1000 bootstrap replications and corresponding two-sided p-values depicted in Table 3 and in (**a**, **b**, **e**, **g**).

administered 3 months after initiation of standard TB treatment. Encouragingly, H56:IC31 elicited robust cellular immune responses with expansion of vaccine antigen-specific CD4 T-cells. Finally, rather than augmenting immunity as hypothesized, etoricoxib treatment did not improve Mtb-specific immunity but rather reduced circulating vaccine-responsive T-cells when combined with H56:IC31-vaccination.

Our findings open up for further investigations with therapeutic vaccines aiming for treatment shortening and/or improving treatment outcomes of both DS-TB and MDR-TB. This points towards a multi-purpose potential for the H56:IC31 candidate that also holds promise regarding prevention of infection and disease progression. Notably, a clinical trial of the subunit candidate M72/AS01E, also tested for its therapeutic potential in TB patients, was prematurely ended due to severe injection site reactions[21]. The most frequent AEs in the

H56:IC31-group were arthralgia, fatigue, and gastro-intestinal symptoms comparable to the other study groups and thus unlikely attributable to systemic vaccine reactions. Still, it is important to note, that the previous listed studies on H56:IC31, as well as the present study, do not have a sample size fitted to assess infrequent AEs and it remains to be determined whether H56:IC31 is efficacious in the human target populations. In that regard, other therapeutic vaccine candidates, based on inactivated mycobacteria, have demonstrated capacity for faster sputum smear conversion, resolution of radiological extent[22,23], and sputum culture conversion[24], which are endpoints that might be explored for H56:IC31 in phase II/III trials.

As hypothesized, immunization with H56:IC31 led to vaccine-specific cellular immune responses significantly above the levels seen in the patients receiving standard TB treatment only. This indicates that pre-existing immune responses during active TB do

not block vaccine boosting of cellular immunity. It is indeed encouraging that we observed comparable frequencies of Ag85B and ESAT-6 specific cytokine+ CD4 T-cells in TB patients after H56:IC31-vaccination to previously reported levels in Quantiferon-TB (QFT) positive healthy adults[16,17]. At two doses H56:IC31 with H56 at 15 μg or 50 μg, respectively, Luabeya et al. report frequencies of Ag85B and ESAT-6 responsive CD4 T-cells in ranges about 0.10–0.40%, peaking after the second vaccination for Ag85B and after the first vaccination for ESAT-6[16,17]. Using the same 5 μg doses of H56 as in our trial, Suliman et al. report similar findings for Ag85B with regard to frequencies and peaking magnitudes after the second H56:IC31 dose[17]. Notably, they reported higher frequencies of ESAT-6 responsive CD4 T-cells (about 0.80%) following the first vaccination, but the levels were reduced and subsequently comparable to our TBCOX2 trial following the second vaccination[16,17]. Although triple-producing IFNγ+IL2+TNFα+ were more predominantly observed in QFT positive healthy subjects[16,17], significant increase were also observed in response to Ag85B in the H56:IC31-group in our trial.

The observed capacity of H56:IC31 to induce IL-2 producing CD4 T-cells (alone or in combination with other cytokines) is of particular interest, considering that this cytokine serve as a proxy for T-cell proliferation and memory[25,26]. Also, CD4 T-cell subsets expressing IL-2 and at least one additional cytokine are specifically suggested as targets for TB vaccine development[19,20]. Within the H56:IC31-group, expansion of IL-2 producing polyfunctional CD4 T-cell subsets was observed both by Fluorospot and WB-ICS evaluation post vaccination although the differences reached significance in post-hoc analyses only for a few subsets. Following the H56:IC31-induced peak, we observed a contraction phase with declining frequencies. Nevertheless, frequencies of circulating Ag85B-specific T-cells remained elevated in the H56:IC31-group at day 238, indicating longevity. Also, our data on H56 antibody responses are comparable to the study by Luabeya et al.[16] where the proportion of H56-IgG responders in QFT positive subjects were 63–86% depending on the doses and number of vaccines given whereas the proportion of H56 IgG responders in our TBCOX2 trial was 75% in both vaccine arms following two administrations.

The registered COX-2i, etoricoxib, with potential beneficial effects on excessive inflammation and T-cell immunity, did not show any major safety concerns in our study. Still, the number of AEs, typically gastrointestinal symptoms, were higher in the etoricoxib groups compared to controls and the H56:IC31-group. These symptoms were reported as possible intervention-related AEs in the groups receiving etoricoxib, but could be explained by already well-characterized side-effects of standard TB treatment. Inspired by findings of improved effector T-cell and humoral immunity by COX-2i therapy in HIV infection[27] and bovine TB[28], we assessed the safety of combining both etoricoxib and H56:IC31 adjunctive to TB treatment. The safety profile was comparable to etoricoxib alone and thus well-known side-effects were not potentiated by H56:IC31. However, opposing our original hypotheses, etoricoxib did not improve cellular or humoral immunity adjunctive to standard TB treatment, but in contrary significantly reduced H56:IC31 specific immunity.

COX-i are widely used drugs for treatment of acute and chronic anti-inflammatory conditions. They are often used to treat TB-related symptoms and side-effects of standard TB treatment like arthralgia and myalgia. Studies on their efficacy in preventing TB-IRIS (NCT02060006) and treating TB meningitis in HIV infection (NCT03927313) are ongoing, and a study of adjunctive ibuprofen in extended drug-resistant (XDR) TB has

recently been completed (NCT02781909). No data from these trials are available yet. Although generated in an open label phase I study with low numbers of patients, our findings question the potential of COX-2i as HDT. Nevertheless, we acknowledge the wide heterogeneity in TB patients and the possibility that COX-i could be beneficial in some patient subgroups. In line with this, reports from mouse models indicate a broad spectrum of outcomes depending on infection kinetics, route of infection, and/or bacterial load. Whereas reduced bacterial load and improved tissue pathology is observed after i.v Mtb infection[5–7], increased bacterial load and impaired T-cell immunity is observed with COX-i-administration after low-dose aerosol Mtb infection[7]. Finally, our study does not rule out potential differences in how NSAIDs, non-selective COX-i and selective COX-2i regulate immune responses in TB patients.

The major limitation of this phase I/II study was the small sample size allowed for each group. This prevented the possibility to evaluate the effect of the interventions on clinical endpoints and the study was therefore based on immunological proxies with an exploratory design and without formal hypothesis testing. Group comparisons resulting in p-values below the decision rule (<0.05), were instead interpreted as hypothesis supporting rather than confirmatory. Also the heterogeneity of patients consisting of both pulmonary and extra-pulmonary TB with various degrees of symptoms calls for a careful interpretation of our data. Further, the effects of H56:IC31 vaccination have in this first-in-human study only been investigated in patients with clinical improvement to standard TB treatment presumably with low bacterial loads. Thus, studies with administration of therapeutic vaccines earlier in the course of standard TB treatment or including TB patients with partial or lacking clinical and/or microbiological response are clearly needed before drawing conclusions regarding the relevance of therapeutic vaccines as HDT. Finally, assessing AEs in a population with TB disease is indeed challenging for three main reasons: (i) TB itself causes symptoms; (ii) Due to the slow progression of disease, patients often fail to notice deterioration until symptoms are reversed by treatment initiation; and (iii) Standard TB treatment is associated with a range of side effects difficult to isolate from eventual intervention-related AEs.

In conclusion, the present study provides essential information for strategic decisions within vaccine development and clinical studies on HDT. Careful selection of promising candidates are required to avoid exhaustion of funding and resources. Although no evident safety issues were observed for either etoricoxib or H56:IC31, our study does not support COX-2i as HDT adjunctive to standard TB treatment. In contrast, optimism regarding the therapeutic potential of the H56:IC31 vaccine is justified as robust expansion of antigen-specific CD4 T-cells were elicited at H56:IC31-administration adjunctive to standard TB treatment. Efficacy studies are warranted to clarify the effect on treatment outcomes and risk of recurrence, for which proven efficacy prepare the grounds for treatment shortening in both DS and MDR TB.

## Methods

**Study design and participants.** The TBCOX2 study was designed as a randomized, open label, controlled, four group (allocation ratio 1:1:1:1), multi-center (Oslo University Hospital and Haukeland University Hospital), safety and explorative phase I/II clinical trial (Fig. 5). Initially, patients aged ≥18 years with confirmed DS-pulmonary TB (GeneXpert MTB/RIF®) without comorbidities (including a negative HIV test), and willing to participate (written informed consent), were included. Due to slow inclusion, a protocol amendment was approved (17.10.2016, Regional Ethics Committee) to include confirmed extra-pulmonary TB (see Supplementary Information, pp. 1–2 for inclusion/exclusion criteria). All participants received standard TB treatment consisting of rifampicin/isoniazid/pyrazinamide/ethambutol as initial two months treatment and thereafter

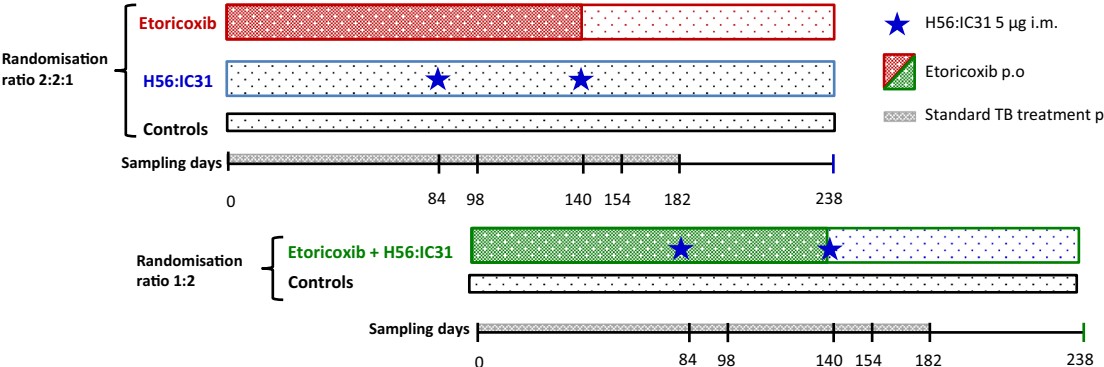

**Fig. 5 Study design.** The TBCOX2 study was designed as a randomized, open label, controlled, four group multi-center phase I/II clinical trial with a final allocation ratio 1:1:1:1 to adjunctive interventions. Target number of participants was 10 per arm for the full analysis set (FAS). The first allocation ratio was 2:2:1:0 to (i) etoricoxib, (ii) H56:IC31 and (iii) controls. The second allocation, initiated following a passed interim analysis of safety (when the last participant in the first allocation had reached study day 98), had a ratio of 0:0:1:2 to (iii) controls and (iv) etoricoxib+H56:IC31. Oral administration (p.o.) of etoricoxib 120 mg was initiated within five days of initiation of standard tuberculosis (TB) treatment (182 days), and continued for 140 days. H56:IC31 5 µg intramuscular (i.m) was administered at day 84 and 140. Samples for immunogenicity analyses were harvested at denoted study days.

rifampicin/isoniazid for a total of 182 days (or longer in settings of single-drug resistance, drug intolerance, or inadequate treatment response).

The protocol was approved by the Regional Ethics Committee (TBCOX2, REK SØ 2015/692) and The Norwegian Medicines Agency (EudraCT Number 2014-004986-26). The study was done in compliance with the Declaration of Helsinki principles and in accordance with the International Conference on Harmonisation's Good Clinical Practices guidelines, and is registered in ClinicalTrials.gov (TBCOX2, NCT02503839).

**Randomization, Interventions, and Procedures.** Participants were enrolled at TB diagnosis and by a computer-generated sequential allocation built into the eCRF software (Viedoc™, Viedoc Technologies AB), randomized to either; etoricoxib, H56:IC31, standard TB treatment only (controls), or etoricoxib+H56:IC31 (Fig. 5). The final treatment allocation to the four study groups was a 1:1:1:1 ratio with a randomization allocation ratio of first 2:2:1:0 and a subsequent randomization allocation ratio of 0:0:1:2. Prior to the second allocation that included the etoricoxib+H56:IC31-group, an interim safety analysis was performed when the last patient in the 2:2:1:0 groups had reached study day 98, according to protocol.

Etoricoxib (Arcoxia®) 120 mg p.o. daily for 140 days (if poorly tolerated reduced to 90 mg) was initiated at the initiation of standard TB treatment (median 0, within a range of 0–5 days of treatment initation). The H56:IC31 vaccine (Statens Serum Institut; SSI, Valneva Austria GmbH) was administered 5 µg intramuscularly at day 84 and day 140 if clinical improvement and two negative sputum examinations by acid fast staining (AFS) or Mtb PCR (in-house) harvested at least seven days apart, were verified. If not, vaccination was postponed until fulfillment of criteria. No placebo interventions were given, thus randomization was open to participants and medical trial investigators. The laboratory technicians and statisticians were blinded to treatment allocation when running laboratory analyses, writing the statistical analysis plan (SAP) and programming statistical analyses.

Demographics, medical history, symptoms, concomitant medication, clinical examination, radiological-, microbiological and routine blood samples were assessed at TB diagnosis (baseline) and during follow-up visits.

**Outcomes and definitions.** The primary outcome was safety and tolerability of etoricoxib and H56:IC31 alone or combined, in patients that received at least one dose of etoricoxib and/or one dose of H56:IC31 (defined as SAS). Safety was assessed by the occurrence of AEs, SAEs, and SUSARs. These outcomes, including incidence of solicited and unsolicited local (injection site) and systemic AE reported for 14 days after vaccination, were assessed by medical trial investigators on days 7, 14, 28, 56, 84, 98, 140, 154, 182, 210, and 238 and included questioning of symptoms, clinical examination, radiology, microbiology, and routine blood sampling. All AEs/SAEs were coded according to the Medical Dictionary for Regulatory Activities (MedDRA) coding system (https://www.meddra.org/) and evaluated for its relationship to the study interventions and severity. As participants had TB at baseline, a deterioration in the FDA toxicity grading scale (mild, moderate, severe; Supplementary Information pp. 3–5) was registered as AEs/SAEs.

We assessed the safety and tolerability by comparing the occurrence of AEs across all study groups during three time periods depending on the intervention given (Fig. 5): from day zero to day 84 we assessed safety of etoricoxib (two groups received adjunctive etoricoxib and two groups received only standard TB treatment during this time interval); from day 85 to day 154 we assessed safety and tolerability of etoricoxib and H56:IC31 alone or combined comparing all four study groups; from day 155 to day 238 we assessed long-term and late-effect safety and tolerability comparing all four study groups (no intervention for any groups during this time interval).

The secondary outcomes were Mtb-specific cellular and humoral immune responses defined and priority ranked a priori depending on the hypothesized impact of the interventions (Supplementary Information pp. 6–7): Hypothesis 1—Etoricoxib administered during the first 84 days of standard TB treatment improves naturally induced Mtb-specific immunity: CD4 T-cell Cytokine+ and IFNγ+ responses to mycobacterial peptides and PPD at the day 0–84 interval were assigned primary priority outcomes. Hypothesis 2—H56:IC31 administered in two doses (day 84 and day 140) elicits vaccine specific immunity. Hypothesis 3—Etoricoxib augments H56:IC31 vaccine-induced responses. Changes in CD4 T-cell Cytokine+, IFNγ+, as well as IgG responses to H56 and individual antigens at day 84–154, were assigned primary priority outcomes for Hypothesis 2 and 3. The secondary outcomes of immunogenicity were assessed in all randomized patients with a valid immunogenicity measurement at baseline and at least one follow-up measurement after randomization (defined as FAS).

**Sample size.** No sample size calculation was done as the TBCOX2 study is an exploratory phase I/II safety study and the first of its kind. Although immunogenicity was not the primary objective of this trial, the target for inclusion, 40 patients, 10 in each study group, was based on results in Mtb-uninfected adults where H56:IC31 elicited significant differences in immunogenicity with 10 patients per group[16].

**Immunogenicity analysis.** Blood samples collected at baseline, day 84, 98, 140, 154, and 238 were analyzed (Fig. 5). Whole blood stimulations were performed on fresh cells subsequently fixed and stored at −145 °C until analysis. All immunogenicity analyses were performed in batches on longitudinally collected frozen samples in a blinded randomized order. See Supplementary Information (pp. 8–10 and Supplementary Fig. 1, p23) for detailed description of the methods; Fluorescence IFNγ/IL-2 immuno-spot (Fluorospot) assay, Whole blood intracellular cytokine staining (WB-ICS) flow cytometry assay and ELISA quantification of anti-H56 IgG in serum.

**Etoricoxib concentration measurements.** Sampling for analyses of etoricoxib concentration was performed at day 14 and day 84. In addition, a 24-h concentration profile was obtained for five participants ~7 days after initiation of etoricoxib (steady state) (Supplementary Information p. 11).

**Statistical methods.** Details on the statistical methods are described in Supplementary appendix (pp. 8–10). All statistical analyses were predefined and detailed in the SAP before data analysis. As the study was planned as an exploratory hypothesis generating phase I/II trial, all group comparisons including p-values were regarded hypothesis supporting and not interpreted as confirmatory. Adjustments for multiple testing were not performed but a hierarchical order of immunogenicity outcomes defined a priori (Supplementary Information pp. 6–7) was judged adequate to assign importance/interpret the results of the respective statistical tests. An increase in immune response was defined as the favorable outcome. Numbers and proportions for categorical variables and medians with

interquartile ranges (IQR) were used to present the baseline characteristics of the participants of the SAS and FAS, respectively.

The primary analyses on safety were performed in the SAS. The safety of etoricoxib and H56:IC31 separately and in combination was assessed based on descriptive tabulations (counts and percentages) of AEs divided into three time periods based on AE start time, day 0 to 84, day 85 to 154 and day 155 to 238.

The secondary outcomes of immunogenicity were assessed in the FAS. We used medians with IQR to present the immunogenicity read-outs at each study time point for descriptive assessments, and numbers and proportions to describe the participants with a > 2-fold increase in IgG antibody response to H56:IC31 from day 84 to any time point after. Immunogenicity read-outs of primary priority for Hypothesis 1 were responses to treatment defined as the change from day 0 to day 84. For Hypothesis 2 and Hypothesis 3, immunogenicity read-outs of primary priority were immune responses to H56:IC31, defined as the change from day 84 to day 154. These readouts were analyzed using median regression with treatment as only independent explanatory factor. The treatment differences were calculated as estimated difference in median response with 95% confidence intervals based on 1000 bootstrap replications and corresponding two-sided p-value. Exploratory post-hoc analyses were performed for Hypothesis 2 on immunogenicity readouts predefined as secondary and tertiary priority from day 84 to day 154 using the same statistical analyses as for the primary priority read-outs. Statistical analyses were performed in STATA version 16 (StataCorp) and figures drawn in GraphPad Prism version 7.02.

**Reporting summary**. Further information on research design is available in the Nature Research Reporting Summary linked to this article.

## Data availability

The authors declare that the data supporting the findings of this study are available in this article and Supplementary Information Files (the original and revised inclusion and exclusion criteria, priority ranking of immunogenicity readouts according to statistical plan, further details on methods, baseline participant characteristics, adherence and safety data, and results of secondary or tertiary priority immunogenicity outcomes). The full source datasets generated during and/or analyzed during the current study are available in the repository of the open science framework (https://osf.io/khvf4).

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

## Acknowledgements

The authors thank the participants in the study, and acknowledge the contributions of referring clinicians at other hospitals, Ingrid Kromann and Peter Andersen, Statens Serum Institut, Denmark for contributing to design and input during the trial and to the manuscript, Helena Strand Clemmensen for helping with the manuscript, clinicians, nurses, and technicians at the Department of Infectious Diseases at Oslo University Hospital and Haukeland University hospital: in particular, Vidar Ormaasen, Mette Sannes, Helene Gjelsås Gabuz, Sarah Nur, Linda Gail Skeie, Ida Wergeland and SATVI for the WB FACS protocol. The study was funded by The Research Council of Norway (RCN, GlobVac no 234493), Oslo University Hospital, Norway, the University of Oslo, Norway, and Statens Serum Institut, Denmark.

## Author contributions

M.R., K.Ta., D.K., and A.M.D.R. designed the study. S.J., K.T., T.M., and A.M.D.R. recruited participants. S.J., K.T., K.R., T.M., P.B., and A.M.D.R. conducted or supervised the clinical and fieldwork. S.J., K.T., K.S., R.M., M.R., M.K., and A.M.D.R. performed or supervised the immunology experiments. C.S.R., I.C.O., and S.J. performed the statistical analysis. S.J., K.T., R.M., M.R., M.K., D.K., T.M., K.Ta., and A.M.D.R. interpreted the data. S.J., K.T., R.M., and A.M.D.R. wrote the manuscript. All authors have full access to the data and reviewed, revised, and gave final approval of the manuscript before submission.

## Competing interests

The authors have no competing interests or other interests that might be perceived to influence the results and/or discussion reported in this paper. M.R., M.K., P.B., and R.M. are employees at SSI that develop the H56:IC31 vaccine. They are not inventors of patents and have no financial interests.
