## [Peer Review File · Nature Communications]

A Phase I/II randomized trial of H56:IC31 vaccination and adjunctive Cyclooxygenase-2-inhibitor treatment in Tuberculosis patientsREVIEWER COMMENTS

Reviewer #1 (Remarks to the Author):

There is recent interest in using new TB vaccine candidates for host-directed therapy. Here the H56:IC31 novel TB vaccine was tested with or without a cyclooxygenase-2-inhibitor, etoricoxib. In this Phase I/II trial 51 individuals were recruited and randomised into four groups, H56:IC31 alone, H56:IC31 with etoricoxib, etoricoxib alone and a control group. It is hoped that a successful therapeutic vaccine could shorten the duration of TB treatment, reduce relapse and also help recovery from drug-resistant TB, as a result of increasing anti-mycobacterial immunity.

The study used etoricoxib, a cyclooxygenase inhibitor (COX-i); previous studies had produced conflicting results regarding the benefit of COX-I in animal models of TB, and the results obtained here did not support the use of etoricoxib as an adjunct to host-directed therapy with a vaccine such as H56:IC31. However the H56 vaccine itself was both safe and immunogenic when given to TB patients on treatment, complementing the earlier results showing it did not give any safety issues in those with Mtb infection rather than disease, or at the end of TB therapy.

Of 222 screened individuals 51 were enrolled and randomised, giving group sizes of 12-15 for the four groups, although in the two groups receiving H56:IC31 the group was then subdivided into those getting the vaccine once and those getting two doses. For the final analyses set the groups ranged from 8-11 and by the final follow up time point at day 238 the groups only contained 6-9 individuals. Other than for severity of disease the groups were well matched. However a very large number of screened individuals were not included due to "breach of inclusion/exclusion criteria". Some indication of these breaches might be useful for readers.

The safety analysis was divided into three time periods, with three SAEs in those receiving etoricoxib from days 0-84, as a result of which the drug was stopped. Between days 85 and 154, one individual who had been vaccinated had a possible IRIS-like reaction AE and was not revaccinated. From day 155-238, there was only one SAE in a patient given etoricoxib.

The immunogenicity analysis used a Fluorospot assay to assess T cell responses to the antigens in H56 and the CFP-10 antigen which served as a useful control as it is not in the H56 antigen. It is encouraging that Ag85B responses were increased, as well as responses to the H56 fusion protein and its constituent peptides, and that these responses were maintained. A whole blood flow cytometry assay also showed increases in T cell cytokine production but these were not significant; this is interesting as it is flow cytometry that has been often used in vaccine trials on the assumption that it was very sensitive. Nevertheless increases in IL-2 producing polyfunctional T cells were observed, which may indicate useful proliferative and memory responses. As an immunologist rather than a trialist, I am not so concerned that some of these analyses were post hoc. Some increase in anti-H56 antibodies was observed but this was not statistically significant. It is worth noting here however that the group sizes were relatively small, and particularly so for those given two rather than one dose of vaccine. The reader may therefore wonder why not all the H56 group were given a second dose of vaccine, or if the intention was to compare one vs two doses, why the group sizes were not increased?

Although the study groups were relatively small, the results did show clearly that the H56 vaccine was safe and immunogenic when given to Tb patients, and that adding etoricoxib to the H56 vaccine decreased rather than increased immunogenicity. The main shortcoming is the small sample sizes in the four groups, but this is acknowledged by the authors.

The methods reveal that the study protocol was amended to enable the recruitment of extrapulmonary

as well as pulmonary TB patients. This must be made clear in the Abstract and in the main text of the results – and provides further heterogeneity within already small group sizes.

Similarly it is only in Figure 1 and when you get to the text of the methods that is clear when vaccination and etoricoxib were given, relative to the start of chemotherapy. It is also pertinent that the H56 vaccine was only given to patients who had had two negative sputum smears and who had shown clinical improvement. Thus the effect of the H56 vaccine has in this study only been investigated in those with low bacterial loads. This might be discussed.

Minor

Line 39, change “totally” to “a total of”.

Line 109, change to “one participant”.

Table 3. In the downloaded pdf copy, the decimal point was replaced by a symbol.

Reviewer #2 (Remarks to the Author):

This manuscript reports a phase IIa randomised open label clinical trial evaluating a subunit vaccine, Cox 2 inhibition, or the two together, in patients with TB disease. The rationale for the trial was that host directed therapy, with either or both of the interventions evaluated here, may be useful as an adjunct to TB treatment.

The numbers are small, and the controls and the etoricoxib / H56-IC31 arms had lower symptom scores than the etoricoxib and H56-IC31 arms alone. However this is not an efficacy trial and the outcomes were safety and immunogenicity.

In general the trial was well conducted and the study well designed and reported. the major limitation, as acknowledged by the authors, is the sample size.

Specific comments:

1. I agree the urticarial rash in the etoricoxib alone group should have been an AE or an SAE but not a SUSAR (it is not unexpected). Could the trial team not have re-classified this AE?
2. Was the patient with IRIS HIV+? How often were participants tested? Was this patient re-tested?
3. How does the immunogenicity of H56-IC31 compare between this study and previous trials in BCG-vaccinated / LTBI subjects? This comparison with previously published data should be included. This point is mentioned in the discussion (line 242-245) but no comment on the magnitude and whether these responses were of a comparable magnitude or not.

Responses to Reviewers

Ref: NCOMMS-21-08822A

A phase I/II randomized trial on Safety and Immunogenicity of the Therapeutic H56:IC31 Vaccine and adjunctive Cyclooxygenase-2-inhibitor in patients with Tuberculosis

Reviewer #1

We thank the reviewer for the comments acknowledging our choice of methods, post-hoc analyzes and that we have discussed the limitation of the small sample sizes in the four study groups.

1. Reviewer comment: *For the final analyses set the groups ranged from 8-11 and by the final follow up time point at day 238 the groups only contained 6-9 individuals. Other than for severity of disease the groups were well matched. However a very large number of screened individuals were not included due to “breach of inclusion/exclusion criteria”. Some indication of these breaches might be useful for readers.*

Authors answer: This is a first-in-human phase I/IIa clinical study with safety as primary readout. The exclusion criteria were therefore strict. To ensure and evaluate safety, we included several exclusion criteria in agreement with the vaccine manufacturer and Norwegian drug-regulation authorities as well as the ethic committee. Thus, patients with co-infections as HIV and viral hepatitis, co-morbidities as gastro-intestinal bleeding disorders, anemia, cardiac disease and hypertension as well as pregnant women were excluded. Also, patients that were expected not to collaborate or to move during the study period were not included. The format of the journal does not allow us to give many details on this in the main text, but all information on inclusion/exclusion criteria are given in **Supplementary Data pp. 1-2**.

2. Reviewer comment: *“...Some increase in anti-H56 antibodies was observed but this was not statistically significant. It is worth noting here however that the group sizes were relatively small, and particularly so for those given two rather than one dose of vaccine. The reader may therefore wonder why not all the H56 group were given a second dose of vaccine, or if the intention was to compare one vs two doses, why the group sizes were not increased?”*

Authors answer: We apologize that this information is not clear, but the intention was to give all patients two doses of the vaccine at day 84 and at day 140 as per protocol (not compare two doses). Only one patients did not receive the second dose due to safety concerns for vaccination as judged by PI (an IRIS like condition occurring in the period following the first vaccination). But as shown in Fig. 2 the reasons for the other study discontinuations were not related to the vaccine, such as voluntary discontinuation, moving abroad, pregnancy and SAE possible related to etoricoxib before vaccination. Several of these events also happened in the last follow-up period in the combined etoricoxib+H56:IC31-group long after the second vaccine was given contributing to lower number of patients only at the end of study at day 238. Thus, missing patients were not replaced with new participants. Still, in the H56:IC31 group 10/12 and in the H56:IC31 + etoricoxib group 8/10 received two vaccine doses which we believe is acceptable (see **Table 2**). This is explained in the original manuscript (**Page 4, from Line 109**), and we have also now stated this more clearly also in **Fig. 2** with referral to the revised **Legends for**

Figure 2 describing number of patients receiving two vaccine doses (**Page 17, Line 477**): ***10 patients received two doses of H56:IC31. ***8 patients received two doses of H56:IC31.*

3. Reviewer comment: *The methods reveal that the study protocol was amended to enable the recruitment of extra-pulmonary as well as pulmonary TB patients. This must be made clear in the Abstract and in the main text of the results – and provides further heterogeneity within already small group sizes.*

Authors answer: We thank for this input. Correctly, the information about the protocol amendment to include confirmed extra-pulmonary TB is given in the Method section. We have now also stated the clinical presentation of the TB patients more clearly in the **abstract (Page 2, Line 37)** as well as in the text under **Results** where more details on the proportions of pulmonary patients in the various study groups are given. **Page 4, Line 97:** *The majority of patients had pulmonary TB (etoricoxib-group 11/13 (85%), H56:IC31-group 12/12 (100%), controls 9/12 (75%) and etoricoxib+H56:IC31-group 7/10 (70%)).*

We have also discussed the impact of this briefly under limitations which brings better context to the results. **Page 11, Line 304:** *Also the heterogeneity of patients consisting of both pulmonary and extra-pulmonary TB with various degrees of symptoms calls for a careful interpretation of our data.*

4. Reviewer comment: *Similarly it is only in Figure 1 and when you get to the text of the methods that is clear when vaccination and etoricoxib were given, relative to the start of chemotherapy.*

Authors answer: We agree that this was not clear and information about the time period for etoricoxib treatment and time points for vaccination are now also given in the **Results** section **Page 4, Line 92.**

5. Reviewer comment: *It is also pertinent that the H56 vaccine was only given to patients who had two negative sputum smears and who had shown clinical improvement. Thus the effect of the H56 vaccine has in this study only been investigated in those with low bacterial loads. This might be discussed.*

Authors answer. Thank you for this important comment. We fully agree that by investigating vaccine effects only in patients responding with clinical improvement to standard chemotherapy we can not conclude on the possible effects for all TB patients. Pursuing this further in future studies is indeed pertinent for the relevance of H56:IC31 as a therapeutic vaccine in general and in MDR/XDR TB in particular. Being a first-in-human phase I/IIa clinical study, we were obliged to prioritize safety with criteria of clinical improvement prior to vaccine administration. In this study we have proven that therapeutic vaccination is safe in patients with clinical improvement. Next steps would be to plan larger studies with co-administration of TB vaccine earlier in the course of standard TB treatment and also include TB patients with only partly or lack of clinical response as in infections with resistant Mtb strains. This is shortly discussed in the revised manuscript under limitations.

Page 11, Line 306: *Further, the effects of the H56:IC31 vaccine has in this first-in-human study only been investigated in patients with clinical improvement to standard TB treatment presumably with low bacterial loads. Thus, studies with administration of therapeutic vaccines earlier in the course of*

standard TB treatment and including TB patients with partial or lacking clinical and/or microbiological response are clearly needed before drawing conclusions regarding the relevance of therapeutic vaccines as HDT.

6. Minor

Line 39, change “totally” to “a total of”. This is corrected.

Line 109, change to “one participant”. This is corrected.

Table 3. In the downloaded pdf copy, the decimal point was replaced by a symbol. This was correct format when submitting the original paper and the Journal editorial office will be informed.

Reviewer #2

We thank the reviewer for this thorough review and appreciate that the reviewer conclude that the trial was well conducted and the study well designed, and also acknowledging that the small numbers of participants is due to the fact that this is a phase I/IIa clinical trial where the outcomes were safety and immunogenicity and not an efficacy trial.

Specific comments:

1. Reviewer comment: *I agree the urticarial rash in the etoricoxib alone group should have been an AE or an SAE but not a SUSAR (it is not unexpected). Could the trial team not have re-classified this AE?*

Authors answer: We discovered this mistake first after data lock and discussed if we should re-classify the SUSAR to an AE as this was obviously a misclassification of unknown reason. However, according to the protocol, the relationship assessment done by the investigator should not be altered by the trial safety group. So we concluded to be transparent and not re-classify and instead explain this misclassification in the text. But we agree with the reviewer in the assessment and if the reviewer or editor would prefer us to re-classify based on this information we will do that.

2. Reviewer comment: *Was the patient with IRIS HIV+? How often were participants tested? Was this patient re-tested?*

Authors answer: All participants, including the patient that developed symptoms compatible with IRIS were HIV tested as part of study screening, and all were HIV negative in accordance with the inclusion/exclusion criteria. This is now stated more clearly in the Method section (**Page 12, Line 331**).

However, since Norway is a low HIV endemic country and the risk of getting HIV infected is very low we did not perform repeated HIV tests per protocol unless newly acquired HIV infection was suspected, which did not happen throughout the study. However, paradoxical worsening of tuberculosis in patients without HIV-co-infection has also previously been reported following chemotherapy (Jarisch–Herxheimer reaction).

3. Reviewer comment: *How does the immunogenicity of H56-IC31 compare between this study and previous trials in BCG-vaccinated / LTBI subjects? This comparison with previously published data should*

be included. This point is mentioned in the discussion (line 242-245) but no comment on the magnitude and whether these responses were of a comparable magnitude or not.

Authors answer: We thank the reviewer for pointing out this important aspect which gives us the opportunity to be more specific of this matter. The manuscript have been adjusted to include the term magnitude justified by the following: Our statement refers to two references representing two H56:IC31 vaccine trials, both conducted in South-Africa and applying the same method to assess T-cell responses, namely the qualified 12h stimulation and WB-ICS assay. Importantly, direct comparisons of immunogenicity results should be interpreted with caution due to inter and intra-assay variabilities in the various laboratories and settings.

In the study by Luabeya et al (1, ref 16 in revised paper), healthy QFT+ and QFT- adults received higher doses of H56 (15ug, 50ug) than in our TBCOX2 trial (5ug), but with equal amount of adjuvant (IC31 500nmol). Magnitudes of total cytokine+ CD4 T-cell responses for Ag85B, ESAT6, and Rv2660c were not specifically reported in Luabeya et al., but from interpretation of figures the magnitudes seem to be about 0.10-0.40% for Ag85B and ESAT6, comparable to results in our trial with median frequencies 0.34% and 0.33% of CD4 T cell populations for Ag85B and ESAT6, respectively. However, frequencies in QFT+ subjects in Luabeya et al. typically peaked 14 days following the second vaccination for Ag85B and 14 days following the first vaccination for ESAT6, whilst we observed similar magnitudes of ESAT6-responsive cytokine+ CD4 T cells 14 days following the first and second vaccination. Rv2660c frequencies were very low in both studies (exact data with median, IQR etc were not given in Luabeya et al, but seem on visual interpretation of the figures to be <0.010 to 0.04% without any clear pattern regarding response to the first or second vaccination. In contrast, we observed median frequencies approaching 0.10% following the second vaccination.

Similar magnitudes is also the case for H56 IgG levels following two administrations in the TBCOX2 trial and three administrations in the Luabeya et al trial. Whereas the proportion of H56-IgG responders in QFT+ subjects were 63% (H56 15ug) and 67% (H56 50ug) after two vaccine administrations, increasing to 86% and 78% after three administrations, the proportion of H56:IgG responders in our TBCOX2 trial was 75% in both vaccine arms following two administrations.

In the second vaccine trial, Suliman et al. (2, ref 17 in revised paper) conducted a dose-optimization study assessing the optimal dosing of H56 in 500nmol of the IC31 adjuvant in QFT- healthy adults randomized to receive 5ug, 15ug or 50ug H56 at two administrations. Following the conclusion of 5ug H56 as the optimal dose, two versus three doses were tested in QFT+ and QFT- healthy adults. The same dose of H56 5ug was used in the TBCOX2 trial. Also in Suliman et al., the magnitudes of total cytokine+ CD4 T-cell responses in QFT+ subjects for Ag85B, were similar to our TBCOX2 trial with a median frequency of Ag85B-responsive Cytokine+ CD4 T cells about 0.25% and with similar peaks 14 days following the first and second vaccination. Notably Suliman et al report a magnitude of ESAT6-responsive CD4 T cells about 0.80% peaking 14 days following the first vaccination, which represent a higher magnitude after the first vaccination, but a comparable magnitude after the second vaccination compared to our trial, reporting peaks of similar magnitudes after the first and second vaccination for ESAT6. H56-IgG data are not published in the Suliman et al study. This is discussed in **Page 9, Line 244-257** and **Page 10, Line 267-271** in the revised manuscript.

1. Luabeya AK, Kagina BM, Tameris MD, et al. First-in-human trial of the post-exposure tuberculosis vaccine H56:IC31 in Mycobacterium tuberculosis infected and non-infected healthy adults. *Vaccine* 2015; **33**(33): 4130-40.
2. Suliman S, Luabeya AKK, Geldenhuys H, et al. Dose Optimization of H56:IC31 Vaccine for Tuberculosis-Endemic Populations. A Double-Blind, Placebo-controlled, Dose-Selection Trial. *American journal of respiratory and critical care medicine* 2019; **199**(2): 220-31.

REVIEWER COMMENTS

Reviewer #1 (Remarks to the Author):

The revised manuscript has satisfactorily addressed the issues raised by the reviewers.

Reviewer #2 (Remarks to the Author):

the authors have satisfactorily addressed the reviewers comments.